# The Mechanism and Regulation of the NLRP3 Inflammasome during Fibrosis

**DOI:** 10.3390/biom12050634

**Published:** 2022-04-26

**Authors:** Carol M. Artlett

**Affiliations:** Department of Microbiology & Immunology, College of Medicine, Drexel University, Philadelphia, PA 19129, USA; cma49@drexel.edu; Tel.: +1-215-991-8585

**Keywords:** NLRP3 inflammasome, fibrosis, IL-1β, IL-18, calcium, potassium, therapeutics

## Abstract

Fibrosis is often the end result of chronic inflammation. It is characterized by the excessive deposition of extracellular matrix. This leads to structural alterations in the tissue, causing permanent damage and organ dysfunction. Depending on the organ it effects, fibrosis can be a serious threat to human life. The molecular mechanism of fibrosis is still not fully understood, but the NLRP3 (NOD-, LRR- and pyrin–domain–containing protein 3) inflammasome appears to play a significant role in the pathogenesis of fibrotic disease. The NLRP3 inflammasome has been the most extensively studied inflammatory pathway to date. It is a crucial component of the innate immune system, and its activation mediates the secretion of interleukin (IL)-1β and IL-18. NLRP3 activation has been strongly linked with fibrosis and drives the differentiation of fibroblasts into myofibroblasts by the chronic upregulation of IL-1β and IL-18 and subsequent autocrine signaling that maintains an activated inflammasome. Both IL-1β and IL-18 are profibrotic, however IL-1β can have antifibrotic capabilities. NLRP3 responds to a plethora of different signals that have a common but unidentified unifying trigger. Even after 20 years of extensive investigation, regulation of the NLRP3 inflammasome is still not completely understood. However, what is known about NLRP3 is that its regulation and activation is complex and not only driven by various activators but controlled by numerous post-translational modifications. More recently, there has been an intensive attempt to discover NLRP3 inhibitors to treat chronic diseases. This review addresses the role of the NLRP3 inflammasome in fibrotic disorders across many different tissues. It discusses the relationships of various NLRP3 activators to fibrosis and covers different therapeutics that have been developed, or are currently in development, that directly target NLRP3 or its downstream products as treatments for fibrotic disorders.

## 1. Introduction

NLRP3 (NOD-, LRR- and pyrin–domain–containing protein 3) is a cytosolic protein that has important implications in the recognition of invading pathogens [1]. When activated, it assembles into a large complex protein aggregate with Apoptosis-Associated Speck-Like Protein Containing a CARD (ASC) and caspase-1 to create the inflammasome. There is one oligomer per cell, and this comprises of seven NLRP3 molecules. When assembled, it is the biggest of all the inflammasomes, being approximately 2 μm in diameter [2]. When activated, it triggers the proteolytic cleavage of pro-caspase-1 into active caspase-1. The pro-inflammatory cytokines interleukin (IL)-1β and IL-18 are synthesized as biologically inactive precursors requiring caspase-1 for their activation. Once activated, caspase-1 proteolytically cleaves pro-IL-1β and pro-IL-18 into their active forms and these cytokines are then secreted where they initiate and perpetuate inflammation. IL-1β amplifies the inflammatory response by recruiting immune cells to the site of the infection, it modulates the adaptive immune response [3], and affects the hypothalamus inducing fever [4]. IL-18 is important in the production of interferon-γ [5] and causes an increase in cytolytic activity of natural killer cells and T cells [6]. The caspase-1 secretome also mediates the release of numerous other proteins, many of which are involved in inflammation, cytoprotection, and tissue repair [7]. Some of these proteins include laminin, annexin A2, matrix metalloproteinase-14, interleukin-1 receptor antagonist, plasminogen activator inhibitor 1, plasminogen activator inhibitor 2, friend leukemia integration-1 transcription factor, macrophage migration inhibitory factor, serpin family B member-1, and the transforming growth factor–beta–induced protein.

Inflammasomes were initially studied in immune cells [8,9], however, they are also found in stromal cells such as epithelial cells [10,11], keratinocytes [12,13], fibroblasts [14], and hepatic stellate cells [15]. These cells are also critical components in the first line of defense against pathogens.

Indeed, it has now been shown that NLRP3 can be activated in a wide variety of different cells, by a wide variety of diverse triggers initiated by microbe-derived pathogen-associated molecular patterns (called PAMPs), or danger-associated molecular patterns (called DAMPs) that are generated by the host cell. In this review, we focus on the activation of the NLRP3 inflammasome and its specific role in fibrotic diseases.

## 2. Inflammasomes and Fibrosis

It is now well proven that the NLRP3 inflammasome (possibly alongside other inflammasomes) plays a significant role in fibrosis, most likely by chronic activation of this inflammatory platform driving the continual release of IL-1β and IL-18 (Figure 1). Fibroblasts are sentinel cells and play a significant role in the integrity of tissues. They sense the microenvironment and react to and modulate the polarization of macrophages. IL-6 and transforming growth factor beta-1 (TGF-β1) are key profibrotic cytokines that can cause tissue resident macrophages to polarize to M2s [16,17,18]. Lipopolysaccharide (LPS)-pretreated fibroblasts secreted cytokines that induced macrophage polarization towards the M1 phenotype [19]. To cause further complications, reciprocal signaling between fibroblasts and macrophages may alter the microenvironment and promote fibrosis [20].

When fibroblasts become activated, they differentiate into myofibroblasts and produce excessive amounts of collagen to aid wound repair. However, if they are activated in the absence of a wound, they can cause fibrosis in the tissues. Like any sentinel cell in the body, they have functional inflammasomes [14].

Gasse et al. [21] demonstrated a significant connection between NLRP3, the IL-1 receptor-MyD88 signaling pathway, and fibrosis. This extensive study showed the central role of IL-1β via engagement with its receptor in driving pulmonary fibrosis. They showed that the production of IL-1β was dependent on the inflammasome. They also went on to prove that direct administration of IL-1β to the lungs of mice mediated tissue destruction causing inflammation and fibrosis, and that this was inhibited with IL-1 receptor antagonist (IL-1RA). Fibrosis is also observed in emphysema and is dependent on the inflammasome [22]. The breakdown of lung tissue by elastase releases uric acid, activating the inflammasome causing IL-1β maturation. These observations were attenuated with IL-1RA [22].

Other studies have now confirmed that the activation of the NLRP3 inflammasome by various DAMPs is associated with fibrosis. For example, bleomycin is a commonly used profibrotic molecule that we employed to study the initiating events of fibrosis. We showed that bleomycin requires the NLRP3 inflammasome to mediate fibrosis in an animal model of systemic sclerosis [14]. Gasse et al. [23] also showed that bleomycin induces the release of uric acid, and this damages cell membranes to activate the NLRP3 inflammasome. They also found that the inflammatory signaling mediated by uric acid release was dependent on IL-1 receptor/MyD88 and suggested that this could be an autocrine loop between the IL-1 receptor and NLRP3 activation during fibrosis. Our studies also implicate an autocrine loop between the IL-1 receptor, NLRP3 and fibrosis in patients with systemic sclerosis [14,24]. In addition to the release of uric acid [23,25,26], bleomycin induces other reactive molecules that can active NLRP3. These include reactive oxygen species (ROS) [27,28,29] and adenosine triphosphate (ATP) [30,31]. Bleomycin also induces perturbations in calcium signaling [32] and this activates NLRP3 (discussed further in Section 3.1).

Our laboratories have reported the role of NLRP3 in fibrotic disease systemic sclerosis/scleroderma [14]. We found the activation of caspase-1 resulted in the increased secretion of collagen by inducing the myofibroblast phenotype. We also found numerous inflammasome genes and associated molecules to be highly upregulated in systemic sclerosis myofibroblasts. These genes included AIM2 (increased 11.8-fold), NLRP5 (8.56-fold), NLRP4 (7.48-fold), NLRP3 (7.02-fold), NLRP12 (6.60-fold), NLRP6 (6.28-fold), NOD2 (6.06-fold), NAIP (3.33-fold) and NLRP9 (2.63-fold). Upregulation in associated factors included CARD6 (13.72-fold), PYDC1 (8.32-fold), pyrin (6.91-fold), CARD18 (6.56-fold) and CASP1 (5.71-fold). We also saw increased expression for IL-1β (7.14-fold), IL-18 (5.02-fold and IL-33 (4.13-fold) genes [14]. This general upregulation suggests that there was a global increase in many of the genes associated with inflammasomes, but they may or may not all contribute to disease pathology. We have not evaluated the contribution of any specific inflammasome to fibrosis in systemic sclerosis myofibroblast cell lines.

Myofibroblasts are the pathogenic cells that drive the increased collagen secretion in tissues during fibrosis. We found that by inhibiting caspase-1, either with the chemical Z-YVAD-FMK or by knocking down gene expression with siRNA, there was a significant reduction in α-smooth muscle actin expression and these cells developed thinner stress fibers [14]. However, our findings directly implicate the role of caspase-1 in the pathogenesis of systemic sclerosis. We further corroborated our findings in NLRP3 and ASC deficient mice. We have since found that the IL-1 receptor plays an integral role in perpetuating inflammasome activation and collagen deposition [24], which suggests that there is autocrine signaling mechanism mediated by IL-1β and/or IL-18 that promotes the profibrotic phenotype in these patients (Figure 1).

Studies on other tissues also demonstrate the role of NLRP3 in fibrosis. The liver is a sensitive site for injury due to pathogens or chemicals. It responds by establishing inflammation that leads to fibrosis. When hepatic stellate cells become activated, they upregulate collagen secretion. Hepatic stellate cells are very similar to myofibroblasts in that they are capable of phagocytosis and antigen presentation. They also express high levels of α-smooth muscle actin stress fibers and they have the ability to migrate [33]. Monosodium urate crystals can activate NLRP3 in hepatic stellate cells and this results in fibrosis. Watanabe et al. [15] demonstrated that these crystals upregulated TGF-β1 and COL1A1, induced the reorganization of actin, and promoted cellular stellation. These features were abrogated in NLRP3-KO and ASC-KO mice which did not show increased TGF-β1 or type I collagen (COL1A1) and had significantly reduced α-smooth muscle actin (SMA).

Other inflammasomes may also play a role in fibrosis, but they have not been studied as extensively as NLRP3. NLRP1 is associated with cardiac fibrosis in rats via the activation of the TGF-β1/SMAD (Mothers Against Decapentaplegic) pathway [34,35]. NLRC5 (NLR Family CARD Domain Containing 5) is upregulated in liver tissues of cirrhosis patients [36], and NLRC5 is increased with carbon tetrachloride and its blockade or knockdown of NLRC5-inhibited liver fibrosis [37]. NLRC5 protein is also elevated in keloid fibroblasts and its knockdown lowers COL1A1, connective tissue growth factor (CTGF), and α-smooth muscle actin expression by altering the TGF-β1/Smad pathway [38]. Other studies on NLRC5 activation of the TGF-β1/Smad pathway have focused on cardiac [39,40] and renal tissues [41]. However, a recent study by Quenum et al. [42] reported that NLRC5-deficient mice responded as efficiently to carbon tetrachloride and induced collagen deposition as wild-type mice. This suggests that carbon tetrachloride may activate another inflammasome in addition to NLRC5.

The DNA sensing inflammasome Absent In Melanoma 2 (AIM2), is also correlated with fibrosis. AIM2 is highly elevated in alveolar macrophages of patients with idiopathic pulmonary fibrosis. Its expression is upregulated by histone deacetylases [43]. Interestingly, the hormone aldosterone enhanced AIM2 activation by double-stranded DNA causing fibrosis in an experimental model of chronic kidney disease. AIM2-deficient mice had lower collagen deposition in the kidney and improved proteinuria levels [44]. Activation of AIM2 in normal human dermal fibroblasts by parvovirus B19 produced proinflammatory and profibrotic gene expression [45].

There are two inflammasomes that are known to be negative regulators of fibrosis. These are NLRC4, which induces IL-1RA and delays the progression of fibrosis [46], and NLRP6. Deficiency in NLRP6 was shown to promote collagen deposition in the liver [47,48].

## 3. Mechanisms of NLRP3 Activation in Fibrosis

The NLRP3 inflammasome is activated by a diverse array of molecules that stimulate its assembly. Its activation is crucial for host defense against fungi, bacteria, viruses, and parasites [49,50,51,52] as it detects PAMPS. The NLRP3 inflammasome also detects DAMPS, comprising of a wide variety of molecules that are structurally and chemically different [53,54]. Many of the stimuli that activate NLRP3 participate in altering the fluctuation of ions into or out of the cell (Figure 1). However, the exact stimulus that NLRP3 is sensing is yet to be found, as it is doubtful that this inflammasome directly senses all these molecules independently. Therefore, it is likely that NLRP3 detects a common cellular event that is induced by these divergent stimuli. While activation of NLRP3 is crucial for host defense, its activation has also been associated with many other chronic diseases, including autoinflammatory diseases [55], gout [56], Alzheimer’s disease [57], diabetes [58,59], and fibrosis [14,21,23].

The assembly of the NLRP3 inflammasome is complex and requires engagement with NIMA (Never In Mitosis Gene A)-Related Kinase 7 (NEK7) and ASC proteins [60,61,62,63,64], positioning caspase-1 in close proximity to another caspase-1 allowing for autocatalytic cleavage (Figure 1). When caspase-1 is processed into its active form, it can cut pro-IL-1β and pro-IL-18 into their active forms and they are secreted. The release of these cytokines can lead to various downstream pathologies that result in many chronic profibrotic diseases. When NLRP3 is in its inactive state, the nucleotide binding site in the NLRP3/NEK7 structure is unavailable and the exchange of Adenosine diphosphate (ADP) to ATP does not occur [65]. This exchange is needed for NLRP3 activation. During cell swelling mediated by influx or efflux of various ions, there is a conformational change in the inactive NLRP3 protein. This suggests that the N-terminal domain of NLRP3 inflammasome is key to its activation because it uniquely responds to specific damage and homeostasis associated with various molecular patterns [65].

Various intracellular and extracellular ions have been shown to activate the inflammasome. But there can be substantial cross-talk between ion fluctuations. It is very unlikely that a sole ion is responsible for NLRP3 activation. Many receptors and molecules that activate NLRP3 modulate more than one type of ion. We discuss below pertinent findings regarding fluctuations in various ions that are involved in the activation of the NLRP3 inflammasome, their cross-talk, and how these ion fluctuations are involved in establishing fibrosis.

### 3.1. Calcium (Ca^2+^) Influx

Ca^2+^ is probably one of the most crucial chemicals for biological life. As an electrolyte, Ca^2+^ plays vital roles in many biochemical processes in cells. It is involved in signal transduction, it is a second messenger, it is a neurotransmitter, it has a role in protein translation in the endoplasmic reticulum, and is required for the proper functioning of mitochondria. The resting concentration of Ca^2+^ in the cellular cytosol is approximately 100 nM and is significantly lower that extracellular concentrations [66,67]. To maintain these low intracellular levels, Ca^2+^ must be actively pumped out of the cytosol into the extracellular space. However, excessive extracellular Ca^2+^ can act as a danger signal, as Ca^2+^ is often found to be elevated at sites of infection [68,69] and where there is necrosis [70]. Thus, elevated extracellular Ca^2+^ levels increase intracellular Ca^2+^ levels in cells surrounding these sites. Ca^2+^ levels are important in maintaining the normal functioning of the cell, and extracellular fluctuations can act as a danger signal that activates NLRP3. Studies have demonstrated that different NLRP3 stimuli induce changes in intracellular Ca^2+^ levels [12,34,35]. These include the NLRP3 activators nigericin [36,37,71] and ATP [38], which raise intracellular Ca^2+^ levels.

Detection of Ca^2+^ occurs through a number of different receptors (Figure 2). These receptors include the Ca^2+^ sensing receptor (CaSR) and G Protein-Coupled Receptor Class C Group 6 Member A (GPCR6A). Rossol et al. [71] showed that NLRP3 was activated through these receptors, as the independent inhibition of CaSR and GPCR6A with Calhex231 or NPS2143, respectively, abrogated IL-1β release. These chemicals had no effect on ATP activation of NLRP3 [71]. Elevated extracellular Ca^2+^ increased cytosolic Ca^2+^ levels [71] and these can be inhibited with the intracellular Ca^2+^ chelator BAPTA-AM (1,2-Bis(2-aminophenoxy)ethane-N,N,N′,N′-tetraacetic acid tetrakis(acetoxymethyl ester). BAPTA-AM inhibited NLRP3 activity and decreased IL-1β release [72]. Furthermore, it has been shown that CaSR on neutrophils induced collagen deposition by fibroblasts via NLRP3 activation in a model of cardiac fibrosis [73].

To help maintain intracellular Ca^2+^ levels there are numerous other channels, including non-voltage activated Ca^2+^ channels, that belong to the transient receptor potential (TRP) superfamily containing TRPC, TRPM, and TRPV channels. All TRPC members are activated by phospholipase C or diacylglycerol (Figure 2). The CRAC channel (Ca^2+^ release activated channel) is another crucial pore that regulates Ca^2+^ levels within the cell. ORAI (ORAI Calcium Release-Activated Calcium Modulator) proteins form the pore of the CRAC channel. ORAI1 interacts with stromal interaction molecule-1 (STIM1) which is an ER transmembrane protein that senses the concentration of Ca^2^*^+^*. When Ca^2+^ levels become low, STIM1 aggregates and interacts with ORAI1, opening the CRAC pore allowing for store-operated Ca^2+^ entry. This process is inhibited by Store-Operated Calcium Entry Associated Regulatory Factor (SARAF) which prevents the interaction of STIM1 with ORAI1. Activation of STIM1 promotes NLRP3 activity. This was demonstrated by the targeted silencing of STIM1 and the subsequent inactivation of NLRP3 [74]. In addition, Murakami et al. [75] showed that activation of the phospholipase C calcium signal transduction pathway is important for NLRP3 activity, and inhibitors of this pathway ameliorated the activation of NLRP3.

Calcium channels are known to play a role in fibrosis. Ross et al. [76] reported increases in expression of ORAI1 in cardiac hypertrophy; however, STIM1 expression was unchanged. Furthermore, using the selective CRAC inhibitor, YM58483, they demonstrated reduced collagen deposition in tissues. Nifedipine is a L-type blocker that is used to inhibit Ca^2+^. It has successfully been used in an animal model against bleomycin-induced lung fibrosis [32]. Efonidipine is a T/L-type calcium channel blocker which showed efficacy in preventing interstitial fibrosis in a chronic unilateral ureteral obstruction model; whereas in this model nifedipine had no effect [77]. As stated above, bleomycin activates NLRP3. Chronic unilateral ureteral obstruction also activates NLRP3 [78].

TGF-β1 is a well-known and extensively studied pro-fibrotic cytokine. It causes oscillations in intracellular Ca^2+^. Spontaneous and periodic oscillations of Ca^2+^ occured in cultured fibroblasts and myofibroblasts, directly correlated with subcellular contractile events [79]. TGF-β1 increased these oscillations and they correlated with COL1A1 and fibronectin expression [80,81]. The TGF-β antagonist, SD208, abolished the Ca^2+^ oscillations [82]. Thus, Ca^2+^ influx is critically important in TGF-β1 signaling [82]. Prostaglandin E2 is anti-fibrotic; its mode of action is to disrupt Ca^2+^ signaling and this decreases fibrotic gene expression [83]. It is likely that these effects are cell dependent. The authors propose that the inhibitory effect of prostaglandin E2 on fibroblasts may work via the E series of prostaglandin receptors (EP_2_ and EP_4_) to increase cyclic AMP. Signaling from these receptors is reportedly anti-fibrotic [84], blunting the Ca^2+^ oscillations caused by TGF-β1 and inhibiting activation of calcium/calmodulin-dependent protein kinase II. This further highlights the crucial role for Ca^2+^ in fibrosis.

TRPC1 is involved in the cell’s response to TGF-β1 [42,43]. It also participates in cell survival, migration [42], differentiation [43], and proliferation [44,45]. It is found on the plasma membrane and in the sarcoendoplasmic reticulum. TRPC1 facilitates epithelial-to-mesenchymal transition and fibroblast-to-myofibroblast transition [46,47,48], and the loss of TRPC1 inhibits these events [48]. TRPC1 depletion prevents TGF-β1-mediated Ca^2+^ influx into cells, thereby preventing fibrosis. Studies have shown that TRPC1 is critical in cardiac remodeling and fibrosis [49], implicating TRPC1 as a potential activator of the inflammasome. In mice, it was shown that inflammasome activation of caspase-11, but not caspase-1, degraded TRPC1. This led to increased secretion of IL-1β without effecting cell death and caused a stronger inflammatory response [85]. This data suggests that calcium has an important role in the regulation of inflammasome activation. In our studies we identified an 18 amino acid cleavage product unique to TRPC1 that is antifibrotic [86,87], however, at this stage we do not know how it is released from TRPC1. Considering these observations, it has been hypothesized that Ca^2+^ channel inhibitors might be effective therapeutics for fibrosis, however, there have been mixed results. Many studies show success in animals [32,88], though this has not translated well in human trials [89,90].

In systemic sclerosis, extracellular Ca^2+^ is an important pathological feature in many patients. These patients often develop calcinosis in the fibrotic tissues, and this presentation is usually seen in the limited subtype of scleroderma. It is painful and is often accompanied by soft tissue swelling and ulcers that are complicated by infections. It leads to deformities, particularly in the hands, causing functional limitations [91]. Deposition of Ca^2+^ in the tissues can be observed in other fibrotic disorders. For example, leaflet thickening accompanied by fibrosis and hardening are early pathological features of calcific aortic valve disease [92]. Ca^2+^ deposition is also associated with fibrosis secondary to other injuries such as burns [93,94], and is found associated with silicosis in the lung [95]. Although not seen in human disease, it is a common feature associated with fibrosis in the mouse model of Duchenne’s muscular dystrophy [96,97]. It is conceivable that this increase in extracellular Ca^2+^ may be one mechanism whereby NLRP3 is activated to drive fibrotic disease in these pathologies.

### 3.2. Potassium (K^+^) Efflux

Manipulation of intracellular and extracellular K^+^ flow has also been used for many years to promote or inhibit NLRP3 activation (Figure 3). Initial studies showed that the depletion of K^+^ in the cytosol with ATP, nigericin, or crystals was sufficient to activate NLRP3, while high levels of external K^+^ blocked this activation [98,99,100]. These fluctuations were specific to NLRP3 because alterations in K^+^ levels do not affect the activation of AIM2 or NLRC4 [98,99]. This suggests that a decrease in intracellular K^+^ levels is crucial for NLRP3 activation, and that K^+^ might be the common trigger that induces the conformational change in NLRP3 causing it to engage with NEK7 [61]. Thus K^+^ may be the universal signal in the cytosol behind the activation of the NLRP3 inflammasome [99]. More recent studies have explored this observation further. Tapia-Abellan et al. [65] identified that the inactive protein structure of NLRP3 favored activation when intracellular concentrations of K^+^ were low. They showed that the domains found between the N-terminal pyrin (PYD) and the NATCH domain were important when intracellular K^+^ levels were low [65]. In contrast to the above observations, it has now been proven that ATP, nigericin, and crystals increase Ca^2+^ levels in the cell [75]. This may then deplete intracellular K^+^ levels causing activation of NLRP3.

While K^+^ efflux is required for NLRP3 activation, it is also needed for NLRC4 activation [101]. This activation is mediated by pore forming toxins such as the type III secretion system. Bacterial flagellin is a ligand for NLRC4 activation but it must be translocated into the cell cytoplasm and requires the type III secretion system for this. Arlehamn et al. [101] showed that extracellular pathogens induce K^+^ efflux activating both NLRP3 and NLRC4 inflammasomes, albeit at different extracellular K^+^ concentrations. Other factors also caused fluctuations of K^+^. Similarly, as in the plasma membrane, lysosomal membranes also have K^+^ ion channels of which two have been identified to date. These are the large-conductance and Ca^2+^-activated potassium channel and the transmembrane protein-175 [102,103]. They are involved in the lysosomal release of K^+^ activating NLRP3 [104,105,106]. The P2X7 purinergic receptor when activated by ATP induces mitochondrial ROS [107], which causes the depletion of K^+^ in the mitochondria and cytosol (Figure 3) where the resulting Ca^2+^ activates the NLRP3 inflammasome [108].

K^+^ fluctuations have also been associated with fibrosis thereby implicating changes in intracellular ion fluxes as predisposing factors, but whether this is a common feature of all fibrotic diseases is yet to be determined. Studies show that NLRP3 activation can be triggered by inhaled nano-particulates causing lung fibrosis. These inhaled nanomaterials can be lodged within the lung and phagocytosed by alveolar macrophages. This induces K^+^ efflux leading to lysosomal damage, oxidative stress, and cell membrane alterations. All these factors activate NLRP3. Zheng et al. [109] demonstrated that airborne particulate matter is sufficient to deplete K^+^ from THP-1 cells. They showed a dose dependent increase in ASC and NLRP3 proteins in THP-1 cells exposed to the fine particles. After phagocytosis, the activated macrophages released IL-1β. In animal studies, they were able to demonstrate that there was a dose dependent increase in IL-1β and TGF-β1 levels in the BAL fluids of exposed mice, and lung histology confirmed the increase in collagen deposition in tissues [109]. These solid results suggests that K^+^ is a critical molecule for NLRP3-mediated collagen deposition.

To further highlight the role of K^+^ in fibrosis in susceptible individuals, mechanical ventilation induced lung damage and pulmonary fibrosis [110,111]. Studies by Liu et al. [60] showed that mechanical ventilation can induce NEK7 assembly with NLRP3 and this is mediated via K^+^ efflux. They were also able to demonstrate that glibenclamide, which an ATP-sensitive K^+^ inhibitor, prevented NEK7-NLRP3 assembly. This Food and Drug Administration (FDA)-approved therapeutic led to less ventilator mediated inflammation and less collagen deposition in the lungs of mice. It also suggests that mechanical stretch caused by the ventilator, might drive intracellular K^+^ efflux, resulting in NLRP3 activation [60].

The initial findings implicated the role of intracellular K^+^ levels, with the suggestion that these were sufficient to promote NLRP3 activation. However, more recently this opinion has changed, because NLRP3 can be activated in the absence of K^+^ efflux. This shows that NLRP3 is detecting something other than intracellular K^+^ levels. More recent studies have shown that chemicals such as CL097 and imiquimod can activate NLRP3 independent of K^+^ efflux. CL097 and imiquimod are small molecules that bind to TLR7, but they also have TLR7-independent activities such as NLRP3 activation. They inhibit mitochondrial activation and cause ROS release and thiol oxidation, which is believed to be the mechanism of NLRP3 activation [112]. However, it cannot be ruled out that they are activating Ca^2+^ influx [71]. NLRP3 activation can be inhibited by ebselen, a glutathione peroxidase mimetic that restores voltage-gated potassium channel function [113], and by pyrrolidine dithiocarbamate [112] which activates the NRF2 pathway.

### 3.3. Calcium-Activated Potassium Channels

Ca^2+^-activated K^+^ channels are K^+^ channels that are gated by Ca^2+^. These channels respond to intracellular Ca^2+^ levels and affect the influx of K^+^ through the plasma membrane. There are eight of these channels known, but it is understood that KCa1.1 and KCa3.1 are proinflammatory and play a significant role in many cellular processes, including adhesion, migration, and invasion [114]. They have been associated with autoimmunity [115,116] and cardiovascular disease [117,118]. These channels are activated by membrane depolarization and cytosolic Ca^2+^ levels. Studies by Schroeder et al. [119] showed that hydroxychloroquine targeted these Ca^2+^ activated K^+^ channels and ameliorated NLRP3 activation. Hydroxychloroquine is a derivative of quinine, and is known to be a K^+^ channel inhibitor. These studies also found that hydroxychloroquine inhibited ATP-induced IL-1β secretion and caspase-1 activation. However, they found that hydroxychloroquine had no effect on cytosolic Ca^2+^ levels induced by ATP [119].

Recently, in studies involving fibroblasts derived from patients with pulmonary fibrosis, KCNMB1 (potassium calcium-activated channel subfamily M regulatory beta subunit 1), which codes for the β-subunit of the large-conductance potassium channel (KCa1.1), was identified as being significantly elevated and its DNA was hypermethylated [120]. These observations were correlated to elevated collagen deposition and pulmonary fibrosis [120]. In addition, the increase in expression of KCa3.1 also promotes fibroblast proliferation in pulmonary fibrosis [121]. Activation of KCa3.1 has been associated with various other fibrotic pathologies including diabetic renal interstitial fibrosis [122], cardiac fibrosis [123], and liver fibrosis [124]. The targeted disruption or pharmacological blockade of KCa3.1 suppressed development of renal fibrosis [125]. The role for this channel in inflammasome activation is further highlighted by recent studies that demonstrate the pharmacological activation of KCa3.1 with paraquat-induced NLRP3 activation, showing that these effects were abolished with the KCa3.1 specific inhibitor TRAM-34 [121].

### 3.4. Other NLRP3 Activating Factors

There are many other factors that active NLRP3 and appear to play a role in fibrosis. Chloride efflux was found to activate NLRP3 and enhance ATP-mediated IL-1β secretion [126]. Confirming this observation, high intracellular chloride levels and chloride channel blockers inhibited NLRP3 activation [126,127]. Chloride efflux via chloride intracellular channel-4 was recently associated with fibrosis in systemic sclerosis and in fibroblasts associated with cancer [128]. Translocation of chloride intracellular channel-4 to the cell surface was found to be dependent [129] on the release of reactive oxygen species (ROS) by mitochondria [130].

It is well-established that mitochondria dysfunction is an upstream activating factor of NLRP3. Mitophagy is a crucial inhibitor of NLRP3 activation, as it removes damaged mitochondria and reduces the mitochondrial release of ROS [131]. However, if mitophagy is inhibited then NLRP3 is activated [131,132]. Indeed, mitochondrial dysfunction also causes the release of mitochondrial DNA, and this is also a critical factor in NLRP3 activation [132].

More recently, oxidative stress has been shown to activate NLRP3 and has the potential to be the universal activator of this inflammatory platform. Oxidative stress is a feature of many chronic diseases, including fibrosis. Many of the chemicals used to induce NLRP3 activation also induce ROS. ROS is a recognised danger signal and can be induced with many of the NLRP3 activators [133,134,135]. It is frequently associated with K^+^ efflux [136] and Ca^2+^ influx [137]. Early studies investigating NLRP3 activation suggested that mitochondrial derived ROS was key [138]. A study found that an increase in ROS induced the dissociation of thioredoxin-interacting protein from its inhibitor thioredoxin, allowing for its engagement to NLRP3 and subsequent activation [139]. Despite the recent interest in the literature regarding NLRP3 and evidence for the role of ROS, it is still not fully clear how NLRP3 is activated due to ROS. A more thorough investigation is crucial for understanding how ROS interacts with K^+^ efflux and/or Ca^2+^ influx to mediate activation of NLRP3, and whether it is a primary or secondary event is unknown.

Nuclear factor erythroid 2-related factor 2 (NRF2) is a transcription factor that regulates the expression of genes containing antioxidant response elements in their promoters. NRF2 helps to regulate NLRP3 activation by inducing the expression of antioxidant genes, helping to limit mitochondrial ROS levels [140]. Activation of the NRF2 pathway also lowers fibrotic mediators and has been shown to be useful for the inhibition of fibrotic pathology in animal models of renal, cardiac, and liver fibrosis [141,142,143]. Inhibiting ROS would be a simple pharmacological target for inhibiting NLRP3 inflammasome assembly. Allicin, a phytochemical extracted from garlic, increased superoxide dismutase, and NRF2/HO-1 anti-oxidative activities. As a result, ROS levels were suppressed, and NLRP3 assembly and activation were inhibited. The net result of this was that it reduced fibrosis [144]. Other metabolites that alter the redox state in the cell also decreased NLRP3 activity. Calcitriol, the active metabolite of vitamin D3, attenuated oxidative stress and reduced NLRP3 activity via the activation of the NRF2-antioxidant signaling pathway [145]. Another example is the metabolite from the Krebs cycle, itaconate, and its derivative, 4-octyl itaconate. Both molecules inhibited NLRP3 inflammasome activation, by impeding the interaction of NLRP3 with NEK7 [146]. This was specific for NLRP3 as these compounds had no effect on AIM2 or NLRC4 activity. These studies were confirmed in mice that were depleted of the *Irg1* gene Irg1 codes for the enzyme that metabolizes cis-aconitate to itaconate and CO_2;_ Irg1-deficient mice had heightened NLRP3 activation [146].

## 4. Regulation of NLRP3 Activation by Post-Translational Modifications

NLRP3 has several post-translational modifications including ubiquitylation, sumoylation and phosphorylation. These modifications regulate the expression and activation of this protein. They have been well studied but not in the direct context of fibrosis. However, it stands to reason that these processes must play a part in fibrosis because NLRP3 must be activated for collagen deposition.

In its inactive state, NLRP3 is ubiquitylated. However, for it to become activated it must be deubiquitylated [147]. There are several proteins that regulate the ubiquitylation and deubiquitylation of NLRP3. The F-box and leucine-rich repeat protein-2 recognizes the tryptophan residue at position 73 on NLRP3 and then targets the lysine-689 residue for ubiquitylation and subsequent degradation of the protein [148]. LPS was shown to increase the half-life of NLRP3 by F-box protein-3 which degrades to F-box and leucine-rich repeat protein-2 [148]. Additional factors also regulate the ubiquitylation of NLRP3. These include the dopamine D1 receptor-induced cAMP that binds to NLRP3. This causes ubiquitylation of the leucine-rich repeat on NLRP3. Another regulator of the ubiquitylation process was found to be TRIM31. TRIM31 is the E3 ubiquitin ligase tripartite motif containing protein 31 which interacts with the PYD motif on NLRP3. This interaction causes ubiquitylation at the lysine-48 residue and proteasomal degradation of NLRP3 [149]. However, TRIM31 is also a negative regulator as it was found that both LPS and IL-1β induced the expression of this protein, which limited NLRP3 activation [149]. Alterations in the NLRP3 ubiquitylation process have been associated with fibrosis. Chronic hepatitis C infections cause liver inflammation that can progress to liver cirrhosis, fibrosis, and carcinoma. The NLRP3 is activated in chronic hepatitis C virus-infected hepatocytes via the deubiquitylation of this protein during infection. The inhibition of deubiquitinases abrogated NLRP3 activation and reduced IL-1β maturation as expected, but intriguingly, it also decreased viral protein and reduced the release of hepatitis C virus from cells [150].

The activation of NLRP3 is also regulated by sumoylation. NLPR3 is sumoylated at numerous sites by the E3 SUMO protein ligase MULI. When NLRP3 is activated, the sumoylation is removed by sentrin-specific proteases-6 and -7. Deficiency in these sentrin-specific proteases reduces NLRP3 inflammasome activation [151]. In contrast, however, the SUMO1 protein catalyzes the sumoylation of NLRP3 at the lysine 204 residue and promotes inflammasome activation, whereas the sentrin-specific protease-3 removes the sumoylation and deactivates NLRP3 [152].

Human NLRP3 is phosphorylated on numerous sites, of which two sites (serine-198 and serine 293) are specifically associated with NLRP3 activation [153,154]. When the serine-5 and tyrosine-861 residues are phosphorylated, they are associated with NLRP3 inactivation [155,156].

## 5. Inflammasome Inhibitors as Therapeutics for Fibrosis

Direct inhibitors of the inflammasome are potential therapeutics for fibrotic disease. Several NLRP3 inhibitors are already available with FDA approval and others are going through clinical trials. In addition, numerous therapeutics target the downstream products of an activated inflammasome, and these have proven successful in various fibrotic scenarios.

### 5.1. Targeting IL-1 to Prevent Fibrosis

The spectrum of diseases caused by IL-1 includes a vast array of hereditary and non-hereditary pathologies, understand of which has benefited from pilot studies blocking IL-1 receptor signaling. These diseases range from systemic to tissue-restricted conditions. Our preliminary findings regarding systemic sclerosis suggest that IL-1β plays a significant role upregulating collagen production during fibrosis. Supporting this notion, the literature is replete with evidence for the role of IL-1 in fibrotic disorders. Furthermore, inhibiting the IL-1 receptor or sequestering IL-1 from the circulation inhibits fibrosis (scarring) in tissues and organs (Table 1).

Activated inflammasomes play a significant role in the development and progression of fibrotic pathologies. This is probably mediated by a feed-forward autocrine loop that maintains activation of NLRP3 and the continuous release of IL-1β and IL-18. Both cytokines are profibrotic in the right context. Although we can appreciate the initiating mechanisms that activate the NLRP3 inflammasome and the resultant release of IL-1β and IL-18, what turns an acute response into a chronic response is not yet fully understood. However, studies have demonstrated that IL-1β can have dichotomous roles in its effects on TGF-β1 expression. Luo et al. [179] showed that short stimulation (minutes) with IL-1β and TGF-β1 prevented SMAD3 phosphorylation and inhibited TGF-β1 downstream signaling pathways. In contrast, however, in the right context both cytokines can compound to promote collagen expression. The long-term exposure of cells to IL-1β and TGF-β1 for 24 h induced SMAD3 phosphorylation and upregulated TGF-β1 signaling [179]. In other studies, it has been shown that long-term exposure of endothelial cells to IL-1β induces the transformation of these cells into myofibroblasts [180] which then synthesize large amounts of collagen. Thus, targeting these cytokines or the inflammasome are rational opportunities for fibrosis therapeutics.

IL-1 increases leukocyte recruitment and induces other profibrotic mediators such as IL-6 [181,182,183], and can upregulate TGF-β1 [184]. TGF-β1 is a key profibrotic growth factor for fibroblasts [185,186,187]. It is increased in systemic sclerosis [188,189,190] and pulmonary fibrosis [191,192]. Interestingly, TGF-β1 was recently shown to increase the expression of NLRP3 and its co-activator NEK7, resulting in the activation of NLRP3, which increased mature caspase-1 and IL-1β [193]. Inhibition of this activated pathway was attained using IL-1RA which downregulated these findings [193].

IL-1 had similar effects on TGF-β1 and matrix production in cultured fibroblasts [184] and induced fibroblast proliferation [194]. IL-1β also induces α-smooth muscle actin, a marker for activated fibroblasts/myofibroblasts [14]. Furthermore, fibroblasts derived from fibrotic tissues demonstrate a greater responsiveness to IL-1 than those derived from normal tissues [195,196]. This suggests that there are increased levels of the IL-1 receptor in these cells, or that they are more sensitive to IL-1. In the context of normal wound healing, IL-1 downregulates TGF-β1 when the wound is closed [197,198,199]. However, the mechanism behind fibrosis is complex, poorly understood and there is no “wound” per se. Furthermore, serum IL-1β levels are not a reliable indicator for the role of this cytokine [200] as the effects are often localized in the tissue at the site of the fibrosis.

Thus, considering the above studies and our findings that fibrosis requires IL-1 [14,24], we firmly believe that fibrosis is an IL-1-mediated disease and we propose that by blocking the IL-1 receptor, fibrosis will be abrogated. However, this cytokine signaling pathway is complex and it is not as simple as adding IL-1 to fibroblasts and expecting collagen to be increased, emulating fibrotic disorders. IL-1 has a dichotomous role in pathology which is underappreciated. In fibrotic diseases, the NLRP3 inflammasome is also activated in what we believe to be a feed-forward mechanism that is mediated by IL-1 (and possibly confounded by IL-18). This signaling mechanism differs from the simple addition of IL-1β to normal fibroblasts in the absence of an activated NLRP3 inflammasome. We speculate that these differences could be due to the differences between myofibroblasts that have an activated inflammasome and normal fibroblasts that do not. Several publications point to this argument and suggest that authors are really investigating the mechanism of normal wound healing and not fibrosis. The recent study by Birnhuber et al. [201] found the administration of IL-1 led to the decreased production of collagen and α-smooth muscle actin in the genetic Fra-2 mouse model of scleroderma. Fra-2 is a transcription factor that upregulates collagen deposition in the tissues. This congenic strain does not need an activated inflammasome for fibrosis, and reveals that Fra-2 most likely acts downstream of NLRP3-mediated inflammatory products. In another study using normal pulmonary fibroblasts, IL-1β decreased collagen, F-actin, and α-smooth muscle actin [202]. IL-1β failed to have profibrotic effects on normal fibroblasts, in contrast to TGF-β1, and was found to antagonize the profibrotic effects of TGF-β1 by downregulating collagen synthesis [197]. In our laboratory we also found that the addition of IL-1β to normal fibroblasts in certain circumstances downregulates collagen synthesis. However, the literature is replete with studies demonstrating that blockade of the IL-1 receptor abrogates fibrosis (Table 1), and suggests that in normal fibroblasts there is a crucial pathway that is missing. Thus, we believe that the missing factor in these studies is likely to be an activated NLRP3 inflammasome. To that effect, the NLRP3 activator nigericin induced fibrogenesis in mice [157] and ATP promoted the expression of TGF-β1 [203].

### 5.2. Targeting IL-18 to Prevent Fibrosis

The activation of NLRP3 also regulates the secretion of IL-18 and evidence is now mounting for the role of this cytokine in fibrosis. IL-18 binding protein (IL-18BP) is the natural inhibitor of the IL-18 receptor and competes for binding with IL-18. IL-18 has been shown to promote fibroblast senescence in pulmonary fibrosis causing the cells to become highly secretory [204]. Studies of the expression of IL-18 and IL-18BP in keloid keratinocyte/keloid fibroblast cocultures showed a significant elevation of bioactive IL-18 whereas IL-18BP levels remained the same [205]. This tips the balance of signaling in favor of IL-18. The blockade of the IL-18 receptor with IL-18BP successfully inhibits fibrosis in human and animal models. In a mouse model of pulmonary fibrosis, the neutralization of the IL-18 receptor with IL-18BP abrogated bleomycin-induced pulmonary fibrosis. It attenuated collagen deposition and decreased TGF-β1 and α-smooth muscle actin expression [206]. In vivo studies of renal fibrosis showed that mice treated with IL-18BP had less fibrosis. They also had lower numbers of myofibroblasts and lower levels of fibronectin and collagens in the tissues [207,208]. Furthermore, IL-18 deficiency protected mice from interstitial fibrosis [209]. IL-18 promotes keloid pathogenesis via epithelial-mesenchymal upregulation [205]. Fibroblasts derived from patients with keloids had increased IL-18 levels, while IL-18BP levels remained the same.

### 5.3. Directly Targeting the NLRP3 Inflammasome

There has been a keen interest in the development of therapeutics that target NLRP3 in an effort to control chronic diseases, especially those that result in fibrosis. Pharmacological inhibition of NLRP3 may be collectively more effective at controlling fibrotic diseases, rather than the selective inhibition of IL-1β or IL-18 (Table 2).

The list in Table 2 is by no means exhaustive. While not discussed in depth here, there are numerous phytochemicals that have been found to inhibit NLRP3 activity and play a role in abrogating fibrosis. Screening natural products for inhibitory NLRP3 molecules to treat fibrosis has recently gained significant interest in the scientific community. In vivo and in vitro studies have shown that natural products, such as terpenoids, phenols, and alkaloids, demonstrate significant inhibitory activity against NLRP3 inflammasome. These have been detailed in an excellent review by Ding et al. [210].

**Table 2 biomolecules-12-00634-t002:** NLRP3 inhibitors and their efficacy on fibrosis.

NLRP3 Inhibitor	Mode of Action	Model of Fibrosis	References
MCC950	Blocks the ATPase domain of NLRP3	Improves liver fibrosis caused by NAFLD and schistosomiasis infections; decreases renal fibrosis induced by cisplatin or diabetes. However, it enhances renal inflammation, injury, and glomerulosclerosis in streptozotocin-induced diabetic mice.	[211,212,213,214,215,216]
Glibenclamide (Glyburide)	Blocks ATP mediated K^+^ channels and prevents ASC aggregation	FDA approved. It ameliorates liver fibrosis caused by inflammation and *Brucella abortus*; has a synergistic effect with dimethyl fumarate; alleviates cardiac inflammation and fibrosis, and bladder and kidney fibrosis.	[163,217,218,219,220]
Parthenolide	Alkylates cysteine residues and inhibits ATPase domain of NLRP3	FDA approved. Has shown efficacy against pulmonary fibrosis, peritoneal fibrosis, and liver fibrosis.	[221,222,223,224]
Tranilast (Rizaben)	Blocks NLRP3-NLRP3 and NLRP3-ASC interactions	FDA approved. Effective against pulmonary fibrosis, fibrosis caused by muscular dystrophy; retards eye fibrosis due to chronic GVHD.	[225,226,227]
Oridonin	Blocks NLRP3-NEK7 interaction	Inhibits bleomycin-induced pulmonary fibrosis, is protective against cardiac hypertrophy. CYD0618 oridonin analog has similar effects against fibrosis. In phase I clinical trials.	[228,229,230,231]
Bay 11-7082	Inhibits NLRP3 ATPase activity	Inhibits ischemia-reperfusion mediated fibrosis, reduces TGF-β1 expression in hepatic stellate cells.	[232,233]
CY-09	Inhibits NLRP3 ATPase activity	Effective against cardiac hypertrophy.	[232]
OLT1177	Inhibits NLRP3 ATPase activity	Lowers active caspase-1 and secreted IL-1β and IL-18. Has not been tested directly in an animal model of fibrosis. In phase 1B clinical trial.	[234]
β-hydroxy-butyrate	Inhibits K^+^ efflux	Protective against doxorubicin-mediated cardiac and microvascular fibrosis, reduces kidney fibrosis. Has been used in clinical trials.	[235,236,237]
VX-765	Blocks caspase-1 activity	Ameliorates diabetes-induced renal fibrosis and abrogates oral submucosal fibrosis mediated by arecoline.	[238,239]
16673-34-0	Induces NLRP3 conformational changes blocking activation	Alleviates tissue damage and fibrosis in obstructed kidneys of unilateral ureteral obstructed mice.	[240]
Crocin	Inhibits NLRP3 expression	Improves renal tissue fibrosis caused by hyperglycemia. It has been used in clinical trials.	[241]
Felodipine (PLENDIL)	Calcium channel inhibitor	FDA approved. Protective and therapeutic effect against bleomycin-induced pulmonary fibrosis in mice.	[88]
Nifedipine (Procardia XL)	Calcium channel inhibitor	FDA approved. Disruption of calcium signaling in fibroblasts and attenuation of bleomycin-induced fibrosis by nifedipine.	[32,89]

Inhibitors that can be orally administered. Many of these inhibitors are currently under development by the pharmaceutical industry or are in pre-clinical development and are yet to enter full clinical trials. They inhibit NLRP3 action by various modalities to prevent its activation (Table 2). Furthermore, many of the recent inhibitors identified have been shown to be phytochemicals. This is an exciting group of natural products that could be therapeutically useful for the treatment of fibrotic disorders.

## 6. Conclusions

In summary, the process of fibrosis includes the activation of myofibroblasts, which causes the excessive deposition of extracellular matrix in tissues. The activation of the NLRP3 inflammasome has been strongly associated with this pathology and appears to drive collagen deposition in an autocrine manner mediated by IL-1β and IL-18. NLRP3 activation and regulation is complex with numerous factors and regulators controlling its activation. Many corroborating studies have discussed the role of this inflammatory platform in driving collagen deposition in a wide range of different tissues. Because NLRP3 has been the most extensively studied inflammasome since its discovery, there has been a concerted effort to identify inhibitors of this inflammasome to treat chronic diseases. While these have been relatively successful, IL-1RA has shown significant effectiveness against fibrosis. However, IL-1RA is a biologic that requires injection. Therefore, a more successful approach would be direct chemical targeting of NLRP3 itself.

## Figures and Tables

**Figure 1 biomolecules-12-00634-f001:**
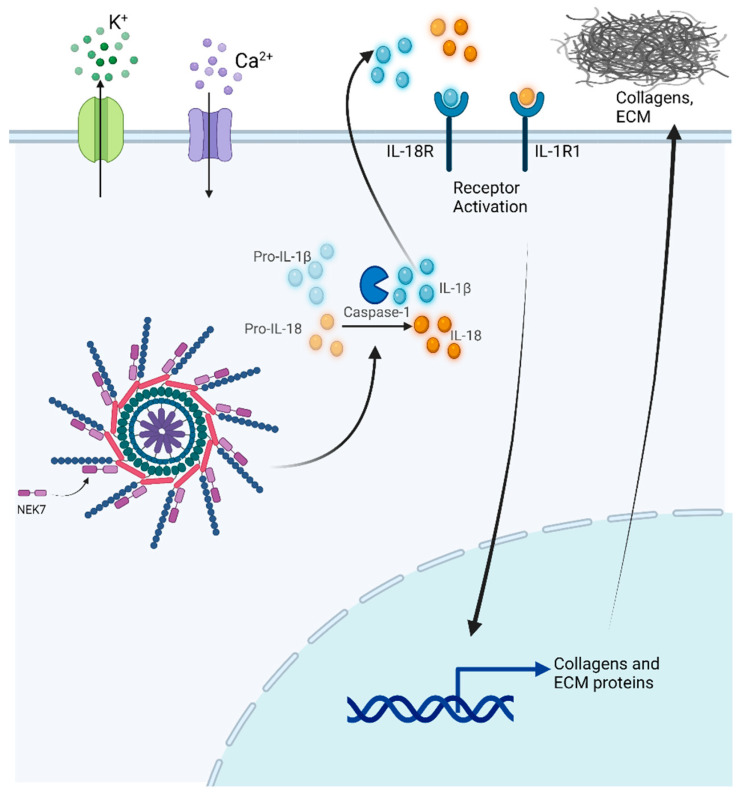
Inflammasome activation drives the expression of collagens and other extracellular matrix proteins. Potassium efflux and calcium influx activate the NLRP3 (NOD-, LRR- and pyrin–domain–containing protein 3) inflammasome. This results in the cleavage and maturation of pro-interleukin (IL)-1β and pro-IL-18 by active caspase-1. When IL-1β and IL-18 are secreted, they can engage their receptors and upregulate the expression of collagen and other extracellular matrix proteins which are released from cells to cause fibrosis. Created using BioRender.com (accessed on 3 March 2022).

**Figure 2 biomolecules-12-00634-f002:**
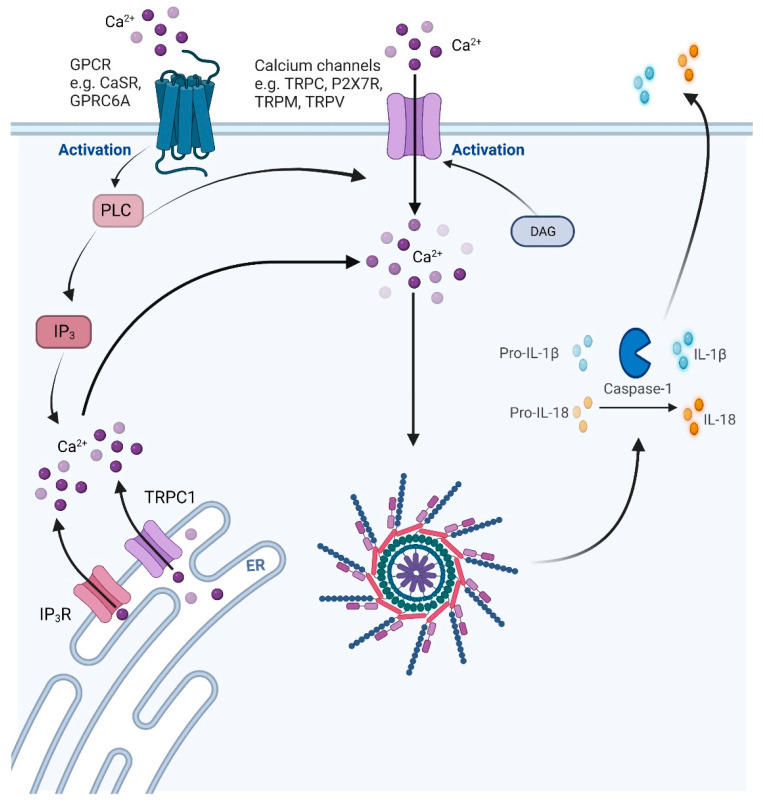
Calcium channels drive the activation of the NLRP3 inflammasome. Extracellular calcium is sensed by G-coupled protein receptors (GPCR) including calcium sensing receptor (CaSR) and G Protein-Coupled Receptor Class C Group 6 Member A (GPCR6A). This upregulates the phospholipase C (PLC) calcium signal transduction pathway which in turn activates inositol 1,4,5-triphosphate (IP_3_) and its receptor (IP_3_R) to release calcium into the cytosol. PLC and diacylglyceral (DAG) also activate other channels including transient receptor potential channels (TRPCs) to allow the influx of calcium into the cell. The influx of calcium into the cytosol activates the assembly of the inflammasome, promoting caspase-1 activation and downstream release of mature IL-1β and IL-18. Created with BioRender.com (accessed on 16 February 2022).

**Figure 3 biomolecules-12-00634-f003:**
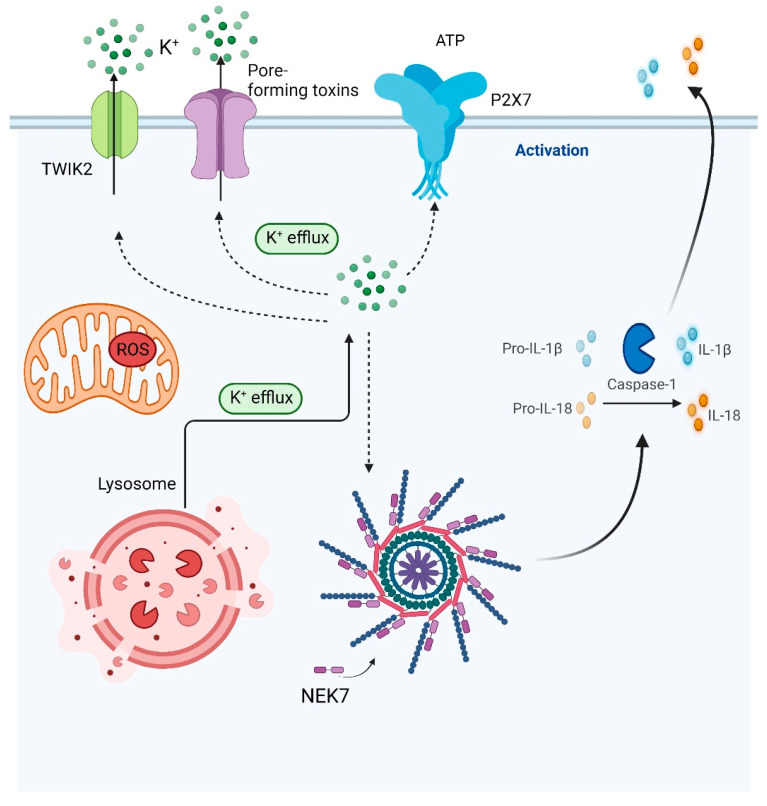
Potassium efflux drives assembly of the NLRP3 inflammasome. TWIK2 and P2X7 are potassium efflux channels that allow the flow of potassium out of the cell in response to ATP. Pore forming channels and the formation of the lysosome also induce potassium efflux. The result of this efflux induces the activation of the NLRP3 inflammasome, most likely through the activation of mitochondrial ROS. Created with BioRender.com (accessed on 17 February 2022).

**Table 1 biomolecules-12-00634-t001:** Animal Models and Human Studies Targeting the IL-1 Receptor.

Type of Fibrosis	Role for IL-1	References
** *Mouse Model* **		
Pulmonary fibrosis	The administration of IL-1β to the lungs promotes inflammation and fibrosis. IL-1RA prevents this. Mice deficient in the IL-1 receptor are resistant to bleomycin-induced fibrosis. SARS-CoV-2 infection in humanized K18-hACE-2 mice treated with anakinra showed less lung fibrosis and reduced mortality.	[21,157,158]
Renal fibrosis	In a rat model of progressive concentric glomerulonephritis, IL-1RA stabilized glomerular injury and reduced interstitial fibrosis. Anakinra (Kineret) reduced renal fibrosis in a mouse model of salt-induced hypertension.	[159,160,161]
Bladder fibrosis	In an animal model of bladder fibrosis, collagen deposition was blocked with Kineret and glyburide. Glyburide inhibited the NLRP3 inflammasome, lowered secreted IL-1β and decreased fibrosis.	[162,163]
Cardiac fibrosis	IL-1 inhibition reduced cardiac fibrosis.	[164]
Skin fibrosis	In two models of scleroderma fibrosis, Anakinra significantly reduced fibrosis. IL-1β is a critical component of radiation-induced skin fibrosis. Inhibiting the IL-1 receptor with IL-1RA decreased fibrotic response in a study of deep incisional wound healing.	[165,166,167]
** *Human Studies* **		
Erdheim-Chester	This is a rare inflammatory disease complicated by retroperitoneal fibrosis. Kineret successfully reduced fibrosis.	[168,169]
Rheumatoid arthritis (RA)	Interstitial lung disease (ILD) is a frequent complication of rheumatoid arthritis. A review of the literature found that most biologics induced ILD in RA patients except for Kineret and hydroxychloroquine. The lifetime risk for developing RA-associated ILD was 7.7%; whereas in the normal population it was 0.9%. In a study of 1346 patients receiving Kineret, only 0.15% developed ILD.	[170,171,172]
Articular arthritis	Kineret was found to prevent joint fibrosis after an anterior cruciate ligament tear, returning the range of motion to normal.	[173,174,175]
Pulmonary fibrosis (PF)	PF secondary to COVID-19 is an expected sequelae in a subset of patients, and can be prevented with anakinra. There were decreased levels of IL-1RA in patients with PF compared to normal individuals, implicating greater IL-1 signaling.	[176,177,178]

## Data Availability

Not applicable.

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
