# Peer review of "The Mechanism and Regulation of the NLRP3 Inflammasome during Fibrosis"

_biomolecules, 2022, doi:10.3390/biom12050634_

Round 1
Reviewer 1 Report
This review encompasses the role of the NLRP3 inflammasome and its activators in fibrotic disorders, and the therapeutic potential of NLRP3 targeting therapy approaches in fibrotic disorders.
The authors handled the subject concisely and fluently. There are only some minor points to be improved.
Comments:
1-In page 5, the authors explained the channels that maintain intracellular Ca2+ levels, such as the TRP superfamily containing TRPC, TRPM, and TRPV channels, the CRAC channel, and ORAI proteins. The involvement of these channels is essential in fibrosis needs to be explained.
2-In page 8, line 293: the authors stated that NLRP3 could be activated independently from K efflux. Instead, molecules such as CL097 and imiquimod inhibit mitochondrial activation and cause ROS release and thiol oxidation leading to the activation of NLRP3 independently of K+ fluxes. Is there any study focused on the role of these molecules in the formation of fibrosis? Or is there any study focused on mitochondria-mediated NLRP3 activation in fibrosis-related diseases?
3- Some minor grammatical errors need to be corrected. Language editing is recommended.
Author Response
1-In page 5, the authors explained the channels that maintain intracellular Ca2+ levels, such as the TRP superfamily containing TRPC, TRPM, and TRPV channels, the CRAC channel, and ORAI proteins. The involvement of these channels is essential in fibrosis needs to be explained. Lines 236-260: this comment has been addressed.
2-In page 8, line 293: the authors stated that NLRP3 could be activated independently from K efflux. Instead, molecules such as CL097 and imiquimod inhibit mitochondrial activation and cause ROS release and thiol oxidation leading to the activation of NLRP3 independently of K+ fluxes. Is there any study focused on the role of these molecules in the formation of fibrosis? Or is there any study focused on mitochondria-mediated NLRP3 activation in fibrosis-related diseases? Lines 418-430: we discuss ROS and fibrosis.
3- Some minor grammatical errors need to be corrected. Language editing is recommended. – these have been corrected.
Reviewer 2 Report
Carol M Artlett elegantly describes the role of NLRP3-inflammasome in fibrosis. After highlighting major outcomes of NLRP3 activation, they provide substantial evidence of its involvement in the process of fibrosis. Then they discuss major mechanisms of activation and regulation via post translational modifications. Finally author describes ways to target this pathway for amelioration of fibrosis-related diseases.
The review appears organic and very informative and summarizes valuable information. Besides many clarifications that I feel need to be provided in order to provide the readership a very clear image of the matter, I am in favor of publication of the manuscript in this journal.
- In the abstract it is not clear what the author means by “spectrum of diseases”.
- It would be helpful to specify in the abstract why IL1 and IL18 are important.
- It would be important to give more general information about other cellular inflammasomes.
- The phrase about “NLRP3 inflammasome is a cytosolic protein” could be reworded, especially since the author mentions after that is a complex of proteins.
- Please specify what other proteins are involved in the caspase-1 secretome.
- It would be important to specify which signals activate NLRP3 in different cells, maybe summarize reported findings in a table.
- More details on how fibroblasts modulate macrophage polarization could be given.
- Authors should better clarify if there is causative relationship specifically between NLRP3 and fibrosis. Are there studies on fibrosis in KO mice for any of the inflammasomes (not just NLRP3) components? Is there any indication of the contribution of the different inflammasomes to fibrosis?
- It is not clear if uric acid is the only, or one of the many signals by which bleomycin activates the inflammasome. It would be important to provide also more details about reported signals, especially in fibrosis and fibrosis-related diseases (although it goes beyond the aim of the review, maybe a summary table could be helpful).
- Author could give more information and details regarding the study with NLRP3 and ASC deficient mice.
- Authors could use abbreviations for DAMPS etc after spelling them entirely the first time they are mentioned.
- It would be important to explain how nigericin and ATP affect calcium. Moreover, it is not clear if author is describing that the sensing of Ca is both direct and indirect. Furthermore, please make more clear if BAPTA decreases NLRP3 activation.
- Please clarify if Calhex231 and NPS2143 do not block ATP-mediated activation of NLRP3.
- The part regarding CRAC and STIM proteins should be integrated with studies about manipulation of these proteins and evaluation of outcome regarding inflammasome.
- It would be important to explain the mechanism by which TGFb causes Ca oscillations.
- Author could specify if it is known the contribution of Ca oscillation due to TGFb in the activation of NLRP3 compared to other signals in the scenarios described.
- How does PGE2 disrupts Ca signalling? Authors could contextualize the role of PGE2 in the sense that many studies suggest role for PGE2 in fibrosis regardless of Ca-mediated activation of NLRP3
- Author could provide evidence for direct connections between NLRP3 and TRPC1. Any findings about causal relationships, mechanisms, regarding interconnection between these?
- The last paragraph of 3.1 could be shortened since information and links to NLRP3 are almost absent.
- It would be helpful to provide info about how ATP, nigericin or crystal decrease K levels.
- It is not clear if the author sustains that K and Ca activate the same downstream mechanisms but K is what drives the process mainly and Ca contribution is a mere consequence of K levels.
- When discussing about pores, it is not clear how NLRP3 vs NLRC4 activation are distinguished regarding K . Moreover please clarify better the words “requirement” vs “prerequisite”.
- Author could provide more details about mechanism behind lysosomal release of K.
- It is not clear if P2X7 acts on Ca levels as well.
- It is not clear how particulate matter depletes K and how mechanical stretch by ventilators drive K efflux.
- Regarding the role of K, from the author arguments it looks like there are strong evidence that K is associated and contributes to inflammasome activation; however authors should highlight how important it is in the process. The studies mentioned indeed, include ligands that can activate NLRP3 in many ways, therefore a definitely line for a specific role of K is missing, especially when author mentions CL097 and imiquimod.
- It would be important to discuss or speculate about the controversy in the literature about hydroxychloroquine and role of K and Ca in its mechanism of action.
- Authors should provide more details about the study in ref 86 (about blockade of KCa3.1). Moreover authors should discuss other studies (if any) involving KCNB1 KO.
- Please define TRX1 and TXNIP.
- Please clarify how ROS are associated with K efflux. Is it causative?
- It would be mandatory to give more information about how ROS can activate mechanistically NLRP3.
- Please provide more details and mechanisms about study in ref 107. Moreover, it would be important if author provided studies for understanding the role of ubiquitination in more close systems; inhibition of ubiquitination in a viral infection may act on multiple mechanisms and not only NLRP3.
- It would be important to add information about redox-modifications of NLRP3. Author should also cite recent literature about metabolites modifying it.
- It is not clear the rationale and history of dual treatments with IL1 and TGFb.
- Please clarify information about ref 151 on how IL1 (produced by M1 macrophages) may induce M2 phenotype.
- Provide reference of IL1 mediating TGFb decrease in “normal” wound healing.
- Correct typos throughout the text.
- Regarding the Fra-2 model, it is not clear if author is suggesting that IL1 in this model drives decrease in collagen because NLPR3 is not involved.
- Author should clarify and speculate about the contrast findings about giving IL1 vs blocking IL1R. First author needs to specify how these treatments are given. Are there reasons to believe that giving IL1 systemically acts on normal fibroblasts while blocking IL1R acts on different cell populations? And how is NLRP3 involved in these 2 scenarios?
- The paragraphs of targeting IL1 and IL 18 could be shortened since they are not strictly involved in the aim of the review.
- Please provide more information about how IL18BP works.
- Authors should speculate on why data about targeting IL18 in fibrosis are more consistent than the ones involving IL1.
- It would be important to give details about at what classes of compounds the ones in table 2 belong to, and if any of them have been used or approved for studies in human or clinical trials. Are there any available compounds targeting Ca-mediated activation of NLRP3?
- It would be mandatory that author spend more words about the natural compounds, about major mechanisms of action and if they have been tested in any pathological condition.
- In the conclusion authors should provide more information about advantages for the use of the chemical inhibitors in the clinic and prospective ideas on how they envision these inhibitors to be administered.
Author Response
- In the abstract it is not clear what the author means by “spectrum of diseases”. – Line 12: this was changed to “…fibrotic disease”
- It would be helpful to specify in the abstract why IL1 and IL18 are important. Line 17: A sentence was added for IL-1β and IL-18.
- It would be important to give more general information about other cellular inflammasomes. Response: This is a review about NLRP3 and fibrosis. Information about other inflammasomes has no place in the abstract.
- The phrase about “NLRP3 inflammasome is a cytosolic protein” could be reworded, especially since the author mentions after that is a complex of proteins. – Lines 32-35: the first two sentences have been rewritten to clarify NLRP3.
- Please specify what other proteins are involved in the caspase-1 secretome. – Line 47-51: additional proteins that are secreted as a result of caspase-1 activity have been listed.
- It would be important to specify which signals activate NLRP3 in different cells, maybe summarize reported findings in a table. Response: While I appreciate this suggestion, this is beyond the scope of the manuscript which is primarily about fibroblasts and NLRP3.
- More details on how fibroblasts modulate macrophage polarization could be given. Lines 74-79: more information has been given.
- Authors should better clarify if there is causative relationship specifically between NLRP3 and fibrosis. Are there studies on fibrosis in KO mice for any of the inflammasomes (not just NLRP3) components? Is there any indication of the contribution of the different inflammasomes to fibrosis? Lines 143-166: I have included information about other inflammasomes.
- It is not clear if uric acid is the only, or one of the many signals by which bleomycin activates the inflammasome. It would be important to provide also more details about reported signals, especially in fibrosis and fibrosis-related diseases (although it goes beyond the aim of the review, maybe a summary table could be helpful). Lines 104-107: I have included additional information about bleomycin
- Author could give more information and details regarding the study with NLRP3 and ASC deficient mice. Lines 141-142: I have given additional information.
- Authors could use abbreviations for DAMPS etc after spelling them entirely the first time they are mentioned. Response: Acknowledged.
- It would be important to explain how nigericin and ATP affect calcium. Moreover, it is not clear if author is describing that the sensing of Ca is both direct and indirect. Furthermore, please make more clear if BAPTA decreases NLRP3 activation. Lines 212 and 222: I have addressed both these comments.
- Please clarify if Calhex231 and NPS2143 do not block ATP-mediated activation of NLRP3. Response: I made this statement in the earlier version of the manuscript “These chemicals had no effect on ATP activation of NLRP3.” I don’t know how to make this statement any clearer.
- The part regarding CRAC and STIM proteins should be integrated with studies about manipulation of these proteins and evaluation of outcome regarding inflammasome. Lines 246-258: I have included additional information.
- It would be important to explain the mechanism by which TGFb causes Ca oscillations. Lines 262-264: I have included additional information about oscillations.
- Author could specify if it is known the contribution of Ca oscillation due to TGFb in the activation of NLRP3 compared to other signals in the scenarios described. Response: This is not known
- How does PGE2 disrupts Ca signalling? Authors could contextualize the role of PGE2 in the sense that many studies suggest role for PGE2 in fibrosis regardless of Ca-mediated activation of NLRP3. Line: 268-272. Additional information is added.
- Author could provide evidence for direct connections between NLRP3 and TRPC1. Any findings about causal relationships, mechanisms, regarding interconnection between these? Lines 280-286: I have included additional information.
- The last paragraph of 3.1 could be shortened since information and links to NLRP3 are almost absent. Line 301-303: I have added a sentence to tie it back to NLRP3.
- It would be helpful to provide info about how ATP, nigericin or crystal decrease K levels. Lines 318-321: information has been added.
- It is not clear if the author sustains that K and Ca activate the same downstream mechanisms but K is what drives the process mainly and Ca contribution is a mere consequence of K levels. Response: Its not clear to any one right now based on the literature. It may be both events are required or one or the other primarily drives NLRP3 activation.
- When discussing about pores, it is not clear how NLRP3 vs NLRC4 activation are distinguished regarding K . Response: I stated in the original version the following “Arlehamn et al. [101], shows that extracellular pathogens induce K+ efflux activating both NLRP3 and NLRC4 inflammasomes, albeit at different extracellular K+ concentrations.” I don’t know how to make this any clearer. Moreover please clarify better the words “requirement” vs “prerequisite”. Line 329: changed to “needed”
- Author could provide more details about mechanism behind lysosomal release of K. Lines 335-338: a sentence was added.
- It is not clear if P2X7 acts on Ca levels as well. Lines 339-341: sentence added.
- It is not clear how particulate matter depletes K and how mechanical stretch by ventilators drive K efflux. Response: The authors of these studies don’t know either. These are their observations.
- Regarding the role of K, from the author arguments it looks like there are strong evidence that K is associated and contributes to inflammasome activation; however authors should highlight how important it is in the process. The studies mentioned indeed, include ligands that can activate NLRP3 in many ways, therefore a definitely line for a specific role of K is missing, especially when author mentions CL097 and imiquimod. Line 372: sentence added.
- It would be important to discuss or speculate about the controversy in the literature about hydroxychloroquine and role of K and Ca in its mechanism of action. Response: This is beyond the scope of this review and the controversies over hydroxychloroquine are irrelevant here.
- Authors should provide more details about the study in ref 86 (about blockade of KCa3.1). Moreover authors should discuss other studies (if any) involving KCNB1 KO. Lines 398-401: additional information added
- Please define TRX1 and TXNIP. Line 425: these have been defined.
- Please clarify how ROS are associated with K efflux. Is it causative? Lines 426-430: several additional sentences expand on this topic
- It would be mandatory to give more information about how ROS can activate mechanistically NLRP3. Replied in point #30.
- Please provide more details and mechanisms about study in ref 107. Moreover, it would be important if author provided studies for understanding the role of ubiquitination in more close systems; inhibition of ubiquitination in a viral infection may act on multiple mechanisms and not only NLRP3. Response: Acting on multiple mechanisms in viral infections is irrelevant especially when the study is looking specifically at the NLRP3 protein.
- It would be important to add information about redox-modifications of NLRP3. Author should also cite recent literature about metabolites modifying it. Lines 441-450: Additional information has been given about other metabolites, etc.
- It is not clear the rationale and history of dual treatments with IL1 and TGFb. Line 517: added a sentence.
- Please clarify information about ref 151 on how IL1 (produced by M1 macrophages) may induce M2 phenotype. Response: the sentence was deleted.
- Provide reference of IL1 mediating TGFb decrease in “normal” wound healing. Line 537: References provided.
- Correct typos throughout the text. Response: corrected
- Regarding the Fra-2 model, it is not clear if author is suggesting that IL1 in this model drives decrease in collagen because NLPR3 is not involved. Response: correct. That is what I say in the original version of the manuscript.
- Author should clarify and speculate about the contrast findings about giving IL1 vs blocking IL1R. First author needs to specify how these treatments are given. Are there reasons to believe that giving IL1 systemically acts on normal fibroblasts while blocking IL1R acts on different cell populations? And how is NLRP3 involved in these 2 scenarios? Response: this was extensively discussed in the original version – Lines 542-570. Giving IL-1 is not used as a therapeutic however giving IL-1RA is.
- The paragraphs of targeting IL1 and IL18 could be shortened since they are not strictly involved in the aim of the review. Response: I have not shortened them. These are paragraphs on NLRP3 inflammasome products and their role in fibrosis. They have a place in this discussion.
- Please provide more information about how IL18BP works. Line 574: I wrote in the original version “IL-18 binding protein (IL-18BP) is the natural inhibitor of the IL-18 receptor and competes for binding with IL-18.”
- Authors should speculate on why data about targeting IL18 in fibrosis are more consistent than the ones involving IL1. Response: I never said it was more consistent. The literature does not support this statement by the reviewer either.
- It would be important to give details about at what classes of compounds the ones in table 2 belong to, and if any of them have been used or approved for studies in human or clinical trials. Are there any available compounds targeting Ca-mediated activation of NLRP3? Response: I do not understand what you mean by classes. In the table I indicated whether the chemicals are FDA approved or in clinical trial if known. I have included calcium inhibitors.
- It would be mandatory that author spend more words about the natural compounds, about major mechanisms of action and if they have been tested in any pathological condition. Response: It is not mandatory. There is an excellent review that I have cited. I wrote in the original version “In vivo and in vitro studies have shown natural products, such as terpenoids, phenols, and alkaloids all demonstrate significant inhibitory activity against NLRP3 inflammasome. They have been covered in an excellent review by Ding et al [211].”
- In the conclusion authors should provide more information about advantages for the use of the chemical inhibitors in the clinic and prospective ideas on how they envision these inhibitors to be administered. Lines 619-621: I have added information requested.